# OpenReview forum: "Shall We Team Up: Exploring Spontaneous Cooperation of Competing LLM Agents"
_ICML.cc/2024/Workshop/Agentic_Markets — Agentic Markets @ ICML'24 Poster_

### Official Review · Reviewer_1p2X · 2024-06-17
**Relevant paper on agent behavior in markets**

**Rating:** 6
**Confidence:** 4

**Review:**

weak points:
- the dataleak prevention is not very well articulated
- would be great if the work can compare with some existing studies on k-lelve reasoning and theory of mind in LLM agents

strong points:
This work studied how LLM agents may engage in spontaneous coordination/collusion using KBC and BC, both are classical game theory examples that can clearly have different results when the agents are LLMs instead of traditional agents, as we are breaking the common knowledge and rationality assumptions. The paper has a solid experiment setup.

other thoughts:
- there is a lot of work lately on what LLMs can do, but foundation models are just unstable, so it would be easy to prompt or engineer them into proving certain points. It would be interesting to frame the study here as studying "what LLMs cannot do"
- simulate llm agents together with trad game theoretic agents competing with each other would give way more convincing results. Current behavior may be the produce of LLMs having an embedded notion of what humans would behave instead of acting in an environment where superhuman calculators exist.